:ᐧᐧᐧ: PLOS | ONE

# Cognitive stress changes the attributes of the three heads of the triceps brachii during muscle fatigue

**Jawad Hussain**[ORCID][1]*, **Kenneth Sundaraj**[1], **Indra Devi Subramaniam**[2]

**1** Centre for Telecommunication Research & Innovation, Fakulti Kejuruteraan Elektronik & Kejuruteraan Komputer, Universiti Teknikal Malaysia Melaka, Durian Tunggal, Malaysia, **2** Centre for Technopreneurship Development, Pusat Bahasa & Pembangunan Insan, Universiti Teknikal Malaysia Melaka, Durian Tunggal, Malaysia

* jawad.hussain@engineer.com

**Data Availability Statement:** The Medical Research and Ethics Committee (MREC) of Malaysia has imposed restrictions on making the underlying data of this study publicly available. The

## Abstract

### Introduction

Cognitive stress (CS) changes the peripheral attributes of a muscle, but its effect on multi-head muscles has not been investigated. The objective of the current research was to investigate the impact of CS on the three heads of the triceps brachii (TB) muscle.

### Methods

Twenty-five young and healthy university students performed a triceps push-down exercise at 45% one repetition maximum (1RM) with and without CS until task failure, and the rate of fatigue (ROF), endurance time (ET) and number of repetitions (NR) for both exercises were analyzed. In addition, the first and last six repetitions of each exercise were considered non-fatiguing (NF) and fatiguing (Fa), respectively, and the root mean square (RMS), mean power frequency (MPF) and median frequency (MDF) for each exercise repetition were evaluated.

### Results

The lateral and long head showed significant differences (P<0.05) in the ROF between the two exercises, and all the heads showed significant (P<0.05) differences in the RMS between the two exercises under NF conditions. Only the long head showed a significant difference (P<0.05) in the MPF and MDF between the two exercises. CS increases the ET (24.74%) and NR (27%) of the exercise. The three heads showed significant differences (P<0.05) in the RMS, MPF and MDF under all exercise conditions.

### Conclusion

A lower ROF was obtained with CS. In addition, the RMS was found to be better approximator of CS, whereas MPF and MDF were more resistant to the effect of CS. The results showed that the three heads worked independently under all conditions, and the non-

data may be provided upon request to the corresponding author or at airehab@utem.edu.my

**Funding:** The authors have no support or funding to report.

**Competing interests:** The authors have declared that no competing interests exist.

**Abbreviations:** 1RM, One Repetition Maximum; CS, Cognitive Stress; CNS, Central Nervous System; ET, Endurance Time; Fa, Fatigue condition; MDF, Median Frequency; MPF, Mean Power Frequency; NCS, No Cognitive Stress; NF, Non-Fatigue condition; NR, Number of Repetitions; RMS, Root Mean Square; ROF, Rate of Fatigue; sEMG, Surface Electromyography; TB, Triceps Brachii.

synergist and synergist head pairs showed similar behavior under Fa conditions. The findings from this study provide additional insights regarding the functioning of each TB head.

## Introduction

The development of muscle fatigue depends on a number of factors, including the exercise type, intensity and duration, the type and fiber composition of the muscle, the synergist pairs, the subject's physical condition and other external factors, such as temperature [1,2,3,4]. Peripheral muscle fatigue can be defined as a decrease in the force-generating capacity of a muscle or group of muscles during or after a particular task [5]. Peripheral muscle fatigue is caused by a loss in the force-generating capacity of motor units (MUs) [6], and to continue performing a desired task, a muscle recruits newer MUs and increases the firing of already recruited MUs. The number of active MUs and the conduction velocity of muscle fibers decreases over time [7], and these effects slow the firing rate of MUs. Eventually, these events lead to the synchronization of MU firing [8], which in turn induces an increase in the surface electromyography (sEMG) root mean square (RMS) and a decrease in the median frequency (MDF). The persistence of such a phenomenon leads to task failure. A previous study revealed that peripheral muscle fatigue is addressed by different motor control strategies depending on the feedback mode [1]. Hence, the involvement of the central nervous system (CNS) in peripheral muscle fatigue cannot be overruled.

The outcome of a type of movement, such as an exercise or day-to-day activity, is affected by the psychological and cognitive demands of the body [9]. Various psychological effects, such as those of thinking about an upcoming assignment or a future meeting, might induce stress [10]. The effect of cognitive stress (CS, also called mental stress or mental load) is more prominent in skeletal muscles closer to the CNS, and the activity in these muscles exhibit a decreasing trend in the caudal direction [11,12]. As described in the literature, the strongest impact of this type of stress is observed in the neck, trapezius and shoulder muscles [11,13]. Increase in CS increases the neuro-motor (also called neural or motor) noise, and this noise propagates in both time and space [14] and might impact muscle performance due to its potential effects on kinematics and joint variability [15]. The MU coordination and control strategy might also be altered by this increased neural noise. As described in a previous study [16], muscle fatigue also increases neural noise. Hence, peripheral fatigue is expected to be affected by an increase in the mental load induced by CS. The effect of CS on fatigue is a complex issue and depends on many factors, including the level of CS. Hence, the behavior of a muscle under fatigue during CS becomes unpredictable due to the neuro-motor noise induced by fatigue and mental stress simultaneously. This variation in the properties of muscles can be observed through sEMG, as shown by a number of researchers [17,18,19].

The triceps brachii (TB), which is the only muscle in the posterior compartment of the arm, acts as an elbow extensor, a horizontal arm abductor, and an antagonist in elbow flexion [20,21]. The anatomical distance of the TB from the CNS renders it more susceptible to the effects of CS compared with distal skeletal muscles, such as the forearm and leg muscles. A previous study [11] analyzed the impact of an increase in the mental load on the TB during finger tapping while the elbow is flexed and observed no change in the activity of the TB, probably due to it being overshadowed by the biceps brachii. Within the TB, the lateral and long heads and the lateral and medial heads act as synergist pairs [22,23], but the behavior of these synergist pairs in the presence of CS has not yet been explored.

CS can be introduced in a number of forms, and its key purpose is to increase the burden on the CNS using the attention diversion technique. For example, multi-tasking is the activity undertaken by an individual who is asked to perform multiple actions at the same time, and previous studies on multi-tasking have employed tests of visual memory (the subject is asked to remember an alphabet or number sequence and reproduce them at a later time), audible memory (the subject is asked to remember a tone and reproduce it at a later time), arithmetic (the subject is asked to solve arithmetic riddles), and logic (the subject is asked to answer logical questions), among others [18,24]. In addition, social [25], work environment [18] and ethical stresses have been commonly used as stress-inducing factors. Based on electroencephalogram (EEG), Zarjam P. classified an arithmetic test into seven levels ranging from a very low to a very high mental load [26]. In the present study, an arithmetic test involving problems with medium to high complexity was used to induce CS in the subjects, and because the subjects were university students, it was expected that they would not find it difficult to answer the arithmetic riddles.

Dynamic contractions are psychologically more demanding than isometric contractions. Cognitively, isometric contractions require focus on postural control, whereas dynamic contractions require both movement and postural control. Thus, the effect of attention diversion techniques is likely to be more easily observable during dynamic contractions. Additionally, dynamic contractions are more closely related to day-to-day activities. Thus, studies on the behavior of a muscle during CS should consider dynamic instead of isometric exercise.

The effect of CS on muscle activity is obvious [27] and can thus be estimated by sEMG. The RMS, which is a temporal parameter, has been used for analyzing variations in sEMG activity due to CS [11] and has been primarily used to determine whether the sEMG activity increases or decreases due to the presence of CS. Although the RMS can be used for the analysis of both isometric [28] and dynamic contractions [29], its application for the latter is not popular [30]. Spectral parameters, such as the mean power frequency (MPF) and median frequency (MDF), have been utilized by some researchers to analyze the impact of CS on sEMG signals [19]. Furthermore, the MPF and MDF tend to decrease with the advent of muscle fatigue and their rate of decrease is termed as rate of fatigue (ROF) [30,31,32,33] and is an important parameter for analyzing the effect of exercise on muscle.

The objective of the current study was to analyze the effects of CS on the three heads of the TB under both non-fatiguing (NF) and fatiguing (Fa) conditions. Because the CS increases neuro-motor noise, it can be hypothesized that the presence of CS would increase muscle activity and that the increases in the activities of synergist muscles could be comparable. Subsequently, it is further hypothesized that the presence of CS would result in a relatively greater decrease in the MPF and MDF.

## Materials and methods

### Participants

Twenty-five young, healthy, recreationally active male university students were recruited for this study. The recruited subjects had no history or on-going diagnosis of neuromuscular disorder in the upper arms. The age of the subjects was 23.8(3.6) years, and their height and weight were 169.1(5.5) cm and 71.2(11.2) kg, respectively. The experimental protocol was approved by the Medical Research and Ethics Committee of Malaysia and is in accordance with the Declaration of Helsinki. The subjects who agreed to participate were informed prior to the experiment and provided written consent. Further, the individual who appears in this manuscript has given written informed consent (as outlined in PLOS consent form) to publish

his/her case details. The experiment was conducted at the university gymnasium, and a physician was available to handle any emergency and aid the researchers.

## Experimental setup

The three heads of the TB were observed via disposable pre-gelled bipolar sEMG electrodes (Kendall™ 100 MediTrace®, Tyco Healthcare Group, USA). The heads were identified with the aid of a physician as described by [34], and based on the SENIAM recommendations, the electrodes were placed on the belly of each head in line with the muscle fibers. The inter-electrode distance was 20 mm, and the skin was shaved, abraded and cleaned prior to electrode application.

A Shimmer 2.0r Model SH-SHIM-KIT-004 instrument (Realtime Technologies Ltd., Ireland) was used to record the sEMG signals. The Shimmer board has built-in filters that capture sEMG data from 5–322 Hz. Each shimmer board was connected to one of the heads of the TB muscle through the corresponding electrode. The reference electrode was placed over an electrically neutral area in the vicinity of the muscle, i.e., lateral epicondyle and olecranon of the shoulder and elbow.

The muscle identification and electrode placements were validated by a physician present on site. The Shimmer system was interfaced with a computer via class 2 Bluetooth®. The raw EMG signals were recorded at a sampling rate of 1 kHz. The computer was placed at a distance of 2 to 3 meters from the subject, and a line of sight was maintained between the subject and computer. The Shimmer Sensing LabVIEW program, accompanying the device was used to store the obtained data on the computer.

## Experimental procedure

When the subject arrived at the site, it was ensured through verbal communication and visual feedback by the physician that the subject was not under any form of CS. The electrodes were placed on the dominant arm of the subject prior to the familiarization session. After placement of the electrodes and sEMG equipment, the subject was asked to warm-up through upper body stretching and exercises consisting of five to eight repetitions using light weights. Subsequently, a resting period of approximately 2 minutes was given.

The subject then stood in-front of triceps push-down machine and held the bar with both hands in the pronated position at shoulder width. During this exercise, the subject kept his arms close to, but not touching, his body and perpendicular to the ground and his torso slightly leaned forward, as shown in Fig 1. The torso was leaned so that hands did not touch the body during full elbow extension. The subject then moved his forearm toward the ground while maintaining the above-described posture until his forearms were fully extended and then slowly moved his forearm back to their starting position (full range of motion (ROM)), and this movement was considered one repetition. The maximal load that each subject held while performing one repetition successfully was termed as one repetition maximum (1RM). It was ensured that the subjects were in the neutral psychological state and any forms of adverse distractions were absent during data collection. Further, the duration of 1RM was conducted at the subject's ease to the same effect. The subject was allowed a 15-minute rest after determination of the 1RM.

Following 1RM test, subjects were asked to perform the TB push-down exercise at 45% 1RM until task failure, under no-cognitive stress (NCS) and cognitive stress (CS). The exercise was performed until exhaustion and the exercise was terminated if the subject could not, control the speed of the bar during the eccentric phase or maintain balance between their dominant and non-dominant arms for two consecutive repetitions. The two exercises were

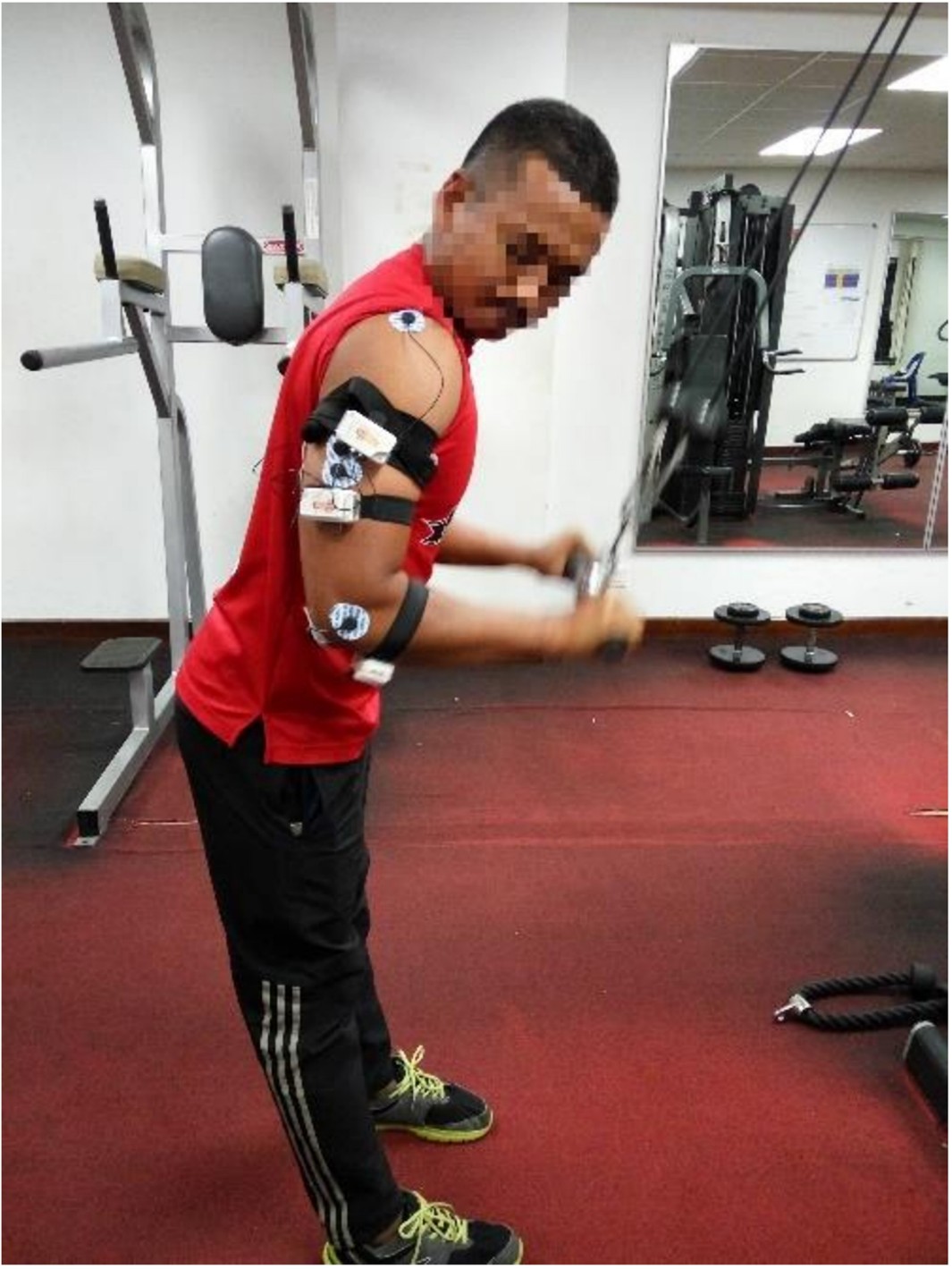

**Fig 1. Posture of the subjects during the triceps push-down exercise.**

randomly assigned to the subject on arrival at the experiment location. For CS, the subject was to solve mathematical additions, while performing the exercise simultaneously, which induce CS in the subject [15,35]. The complexity of the mathematical riddles ranged from medium to high on the scale introduced by [26]. Specifically, the subject was shown two numbers (each

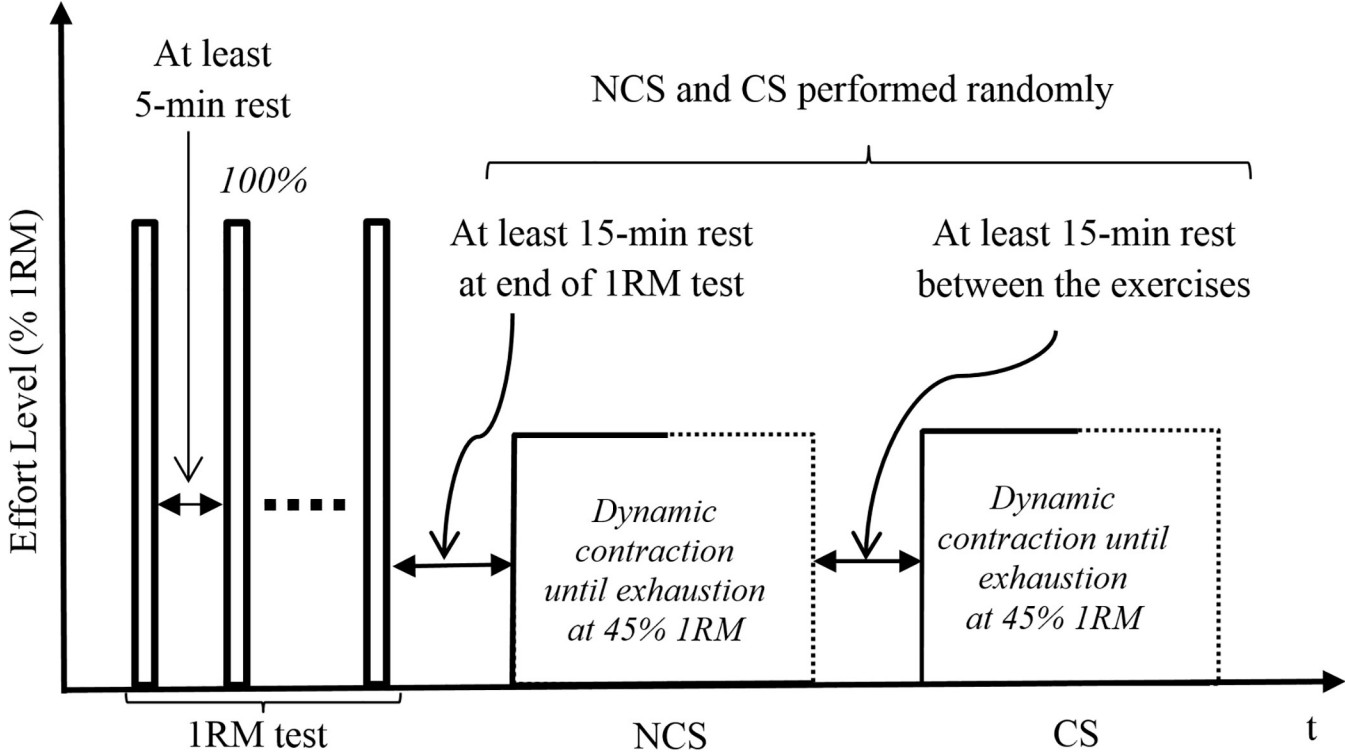

**Fig 2. Temporal experimental procedure for the experiment.**

consisting two to three digits depending on the complexity of the problem) on a computer screen (17″ LCD) that was placed such that the subject could clearly see the numbers. The numbers were displayed in *Times New Roman* font in the middle of the screen for 3 seconds, and the subject was audio alerted prior to their display. The numbers were then removed for a gap time of 2 seconds to test the subject's retention, and a multiple-choice menu with five choices–*a*, *b*, *c*, *d* and *e*, was then displayed for 2 seconds. The subject announced his choice, and 2 seconds later, the next question appeared. For NCS condition, subject simply performed the TB push-down exercise at 45% 1RM without performing any other cognitive activity. A rest period of around 15 minutes was taken between the two exercises. Fig 2 demonstrates the temporal experimental procedure. We note that performing these exercises at 45% 1RM may incur low frequency fatigue (LFF) in the subjects as they may get 55% fatigued from their base value. Although [36] suggests that subjects who get fatigued at 35% or greater from their base value could induce LFF, which could delay the recovery process, the authors reported that after 5 minutes of rest from isometric contractions, muscles regained about 85% of their strength. In addition, [37] commented that LFF is more prominent in eccentric and isometric exercises as compared to dynamic exercises. Hence, we believe that the rest period of about 15 minutes observed in our experiment is enough for the subjects to recover.

If a subject was frequently unable to maintain correct posture i.e. torso leaned too much, subject's torso got straightened or arm abduction varied, a new subject was recruited as a replacement. Our pilot study showed that it is very difficult for a subject to maintain a pre-determined exercise tempo, by either following metronome beeps or visual feedback, and answer the questions simultaneously. Hence, the tempo of the exercise was maintained at subject's ease. Subjects were not allowed to pause between transitions from the eccentric to concentric or concentric to eccentric phases. A custom-made program in LabVIEW measured the

duration of a complete rep from the real-time sEMG data, and it was ensured that all repetitions were within ±15% of this duration. During the experiment, the subject was continuously given verbal encouragement (not amounting to a form of distraction) to exert maximal effort and maintain the tempo. Posture (including all appropriate joint angles) throughout the ROM and exercises was monitored by a dedicated assistant present on site and it was ensured that the subject did not use his body weight to move the bar. Both the exercises were performed on the same day as psychological conditions of subjects may change over different days. The experiment was designed to vary the cognitive stress while keeping all other conditions same.

### Data analyses

The collected data were stored in a computer system for further analysis. Custom-written programs in MATLAB 17 (MathWorks Inc., USA) were used to filter, normalize and evaluate the parameters. A fourth-order bandpass Butterworth filter (cut-off frequencies of 5–450 Hz) was used for filtration. A 512-point Short-Time Fourier Transform (STFT) computed with 50% window overlap was used to estimate MPF and MDF. The filtered and rectified sEMG signals from each exercise and subject were used to extract segments corresponding to the active phase (concentric and eccentric). The active phases were identified by using a 256 ms moving window to obtain the mean of the signal, with a threshold set at 15% of the maximum value in the entire recording, as shown in Fig 3. The RMS, MPF and MDF were calculated for each active phase. Subsequently, the RMS was normalized against a dynamic contraction rather than using a maximal voluntary contraction (MVC), by considering the average RMS amplitude from 1RM repetition. This was done due to the difficulty in finding the optimal joint angle that provides maximal output effort for all three heads. A similar approach was also observed to be used by [38]. The time from the start of the exercise until task failure (endurance time (ET)) and the total number of segments (number of repetitions (NR)) were compared. The first six (non-fatigue) and last six segments (fatigue) were identified, and the MPF, MDF and normalized RMS (onwards RMS) were calculated for all three heads and for all segments. The ROF was calculated from the slop of MPF using regression analysis as suggested by [30,31,32,33].

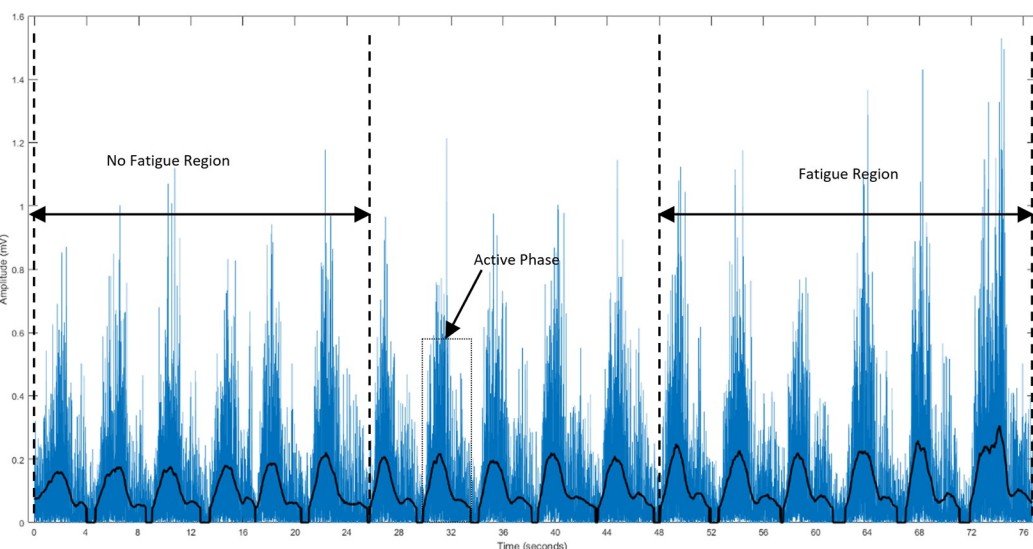

**Fig 3. Filtered and rectified sEMG signal from the lateral head of the TB of a subject.** The active phase, NF and Fa regions are shown.

While both MPF and MDF have been used to explain similar interpretations of physiological aspects of exercises, in this work, both MPF and MDF have been used. This is because several researchers found that there may be potential differences between the two parameters [39,40]. It was further stated by [41] that "certain factors have a differential influence on the different spectral statistics". Hence, we preferred to use both spectral indices.

### Statistical analyses

For each subject, the ROF, ET and NR were obtained for both NCS and CS exercises for all the heads. Subsequently, the RMS, MPF and MDF values were obtained from the active phase under non-fatigue (NF) and fatigue (Fa) conditions in all the heads for each exercise. All data were tested for normality using the Shapiro-Wilks test and found normally distributed. Descriptive statistics was then used to compare ET and NR for the two exercises. One-way repeated measures ANOVA was employed to make comparisons for, ROF between the two exercises in a particular head and ROF between three heads in a particular exercise. Two-way repeated measures ANOVA was employed to analyze the main effects of fatigue-cognitive stress interaction for each head of the TB. For the cases that violated the sphericity assumption, Greenhouse-Geisser corrections were used. Bonferroni adjustments were applied for the post hoc analysis to observe the behavior of parameters among pairs of TB heads. A difference was considered significant if P<0.05, and thus, a confidence level of 95% was considered. The statistical analyses were performed using IBM SPSS 20.0 (SPSS Inc., USA).

### Results

All subjects, with the exception of one, showed significant increase in ET from 58.33(10.89) sec to 82.8(14.83) sec and NR from 21.39(4.54) to 29.67(5.31), in the CS exercise compared with the NCS exercise. The increment was 24.74% and 27% for ET and NR respectively.

Overall, the two exercises showed significant difference (P<0.05) for ROF with mean ROF higher in the NCS exercise in all three heads when compared to the CS exercise. Comparison among the exercises revealed ROF to be statistically significant (P<0.05) for the lateral and long heads only. Fig 4 presents the mean ROF of the three heads.

Table 1 and Fig 5 summarizes the statistical results (one-way ANOVA) of RMS, MPF and MDF under NF and Fa conditions between NCS and CS exercises. Under the NF condition, all three heads showed higher RMS (P<0.05) in CS, while only the long head showed higher MPF and MDF (P<0.05) in NCS. In the Fa condition, both the lateral and long heads showed higher MPF and MDF (P<0.05) in CS. Comparisons between NF and Fa conditions were significantly different (P<0.05) for both exercises in all the heads for all observed parameters, with the exception of RMS in the long head during NCS. Overall, RMS was observed to be higher in Fa condition whereas MPF and MDF were higher in NF conditions.

Table 2 summarizes the statistical results (one-way ANOVA) between the three heads for RMS, MPF and MDF for all four conditions, i.e., NF-NCS, Fa-NCS, NF-CS and Fa-CS. The three heads were statistically significant (P<0.05) for all conditions and parameters with the exception of RMS in NF-NCS. The lateral head produced the highest mean MPF, MDF and RMS in both exercises. The post hoc analysis revealed that only the lateral-long and lateral-medial synergist pairs observed statistical significance under all conditions and parameters (P<0.05). Nevertheless, the long-medial muscle pair showed statistical significance (P<0.05) for MPF and MDF during Fa condition for both exercises.

Interaction results (two-way ANOVA) between fatiguing condition and cognitive stress in all the heads for RMS, MPF and MDF are presented in Table 3. The main effects of CS were found significant in all heads for RMS only, while the fatiguing condition was significant in all

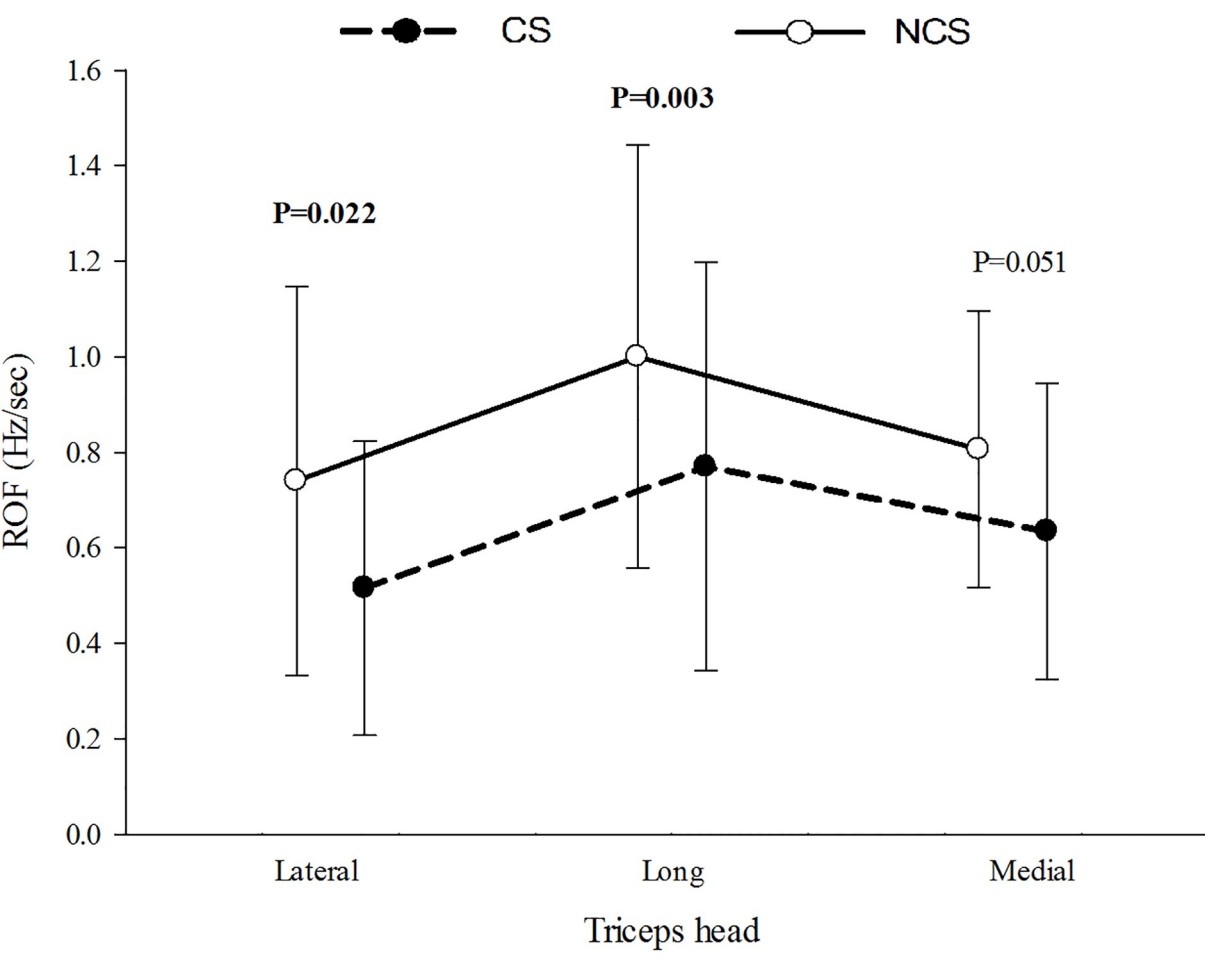

**Fig 4. ROF in µ(SD) of all the subjects during exercise with and without cognitive stress.**

the heads for RMS, MPF and MDF. The interaction between cognitive stress and fatigue was significant in all the heads for RMS and MPF.

Table 4 presents the percentage changes in the RMS, MPF and MDF from NF to Fa conditions in both exercises. Percentage change in RMS was significantly higher in the CS exercise. However, the spectral parameters showed comparable percentage change from NF to Fa

**Table 1. Results of one-way ANOVA for RMS, MPF and MDF between different exercise conditions in each head (P-values).**

| RMS Lat | RMS Lo | RMS Med | MPF Lat | MPF Lo | MPF Med | MDF Lat | MDF Lo | MDF Med |
|---------|--------|---------|---------|--------|---------|---------|--------|---------|
| | | | Between NF-NCS and NF-CS | | | | | |
| <**0.001** | <**0.001** | <**0.001** | 0.34 | **0.004** | 0.057 | 0.92 | **0.002** | 0.17 |
| | | | Between Fa-NCS and Fa-CS | | | | | |
| 0.27 | 0.58 | **0.041** | <**0.001** | <**0.001** | 0.14 | **0.006** | <**0.001** | 0.32 |
| | | | Between NF-NCS and Fa-NCS | | | | | |
| <**0.001** | 0.08 | **0.02** | <**0.001** | <**0.001** | <**0.001** | <**0.001** | <**0.001** | <**0.001** |
| | | | Between NF-CS and Fa-CS | | | | | |
| <**0.001** | <**0.001** | <**0.001** | <**0.001** | <**0.001** | <**0.001** | <**0.001** | <**0.001** | <**0.001** |

*****Bold** font indicates statistically significant. Lat–Lateral, Med–Medial and Lo–Long.

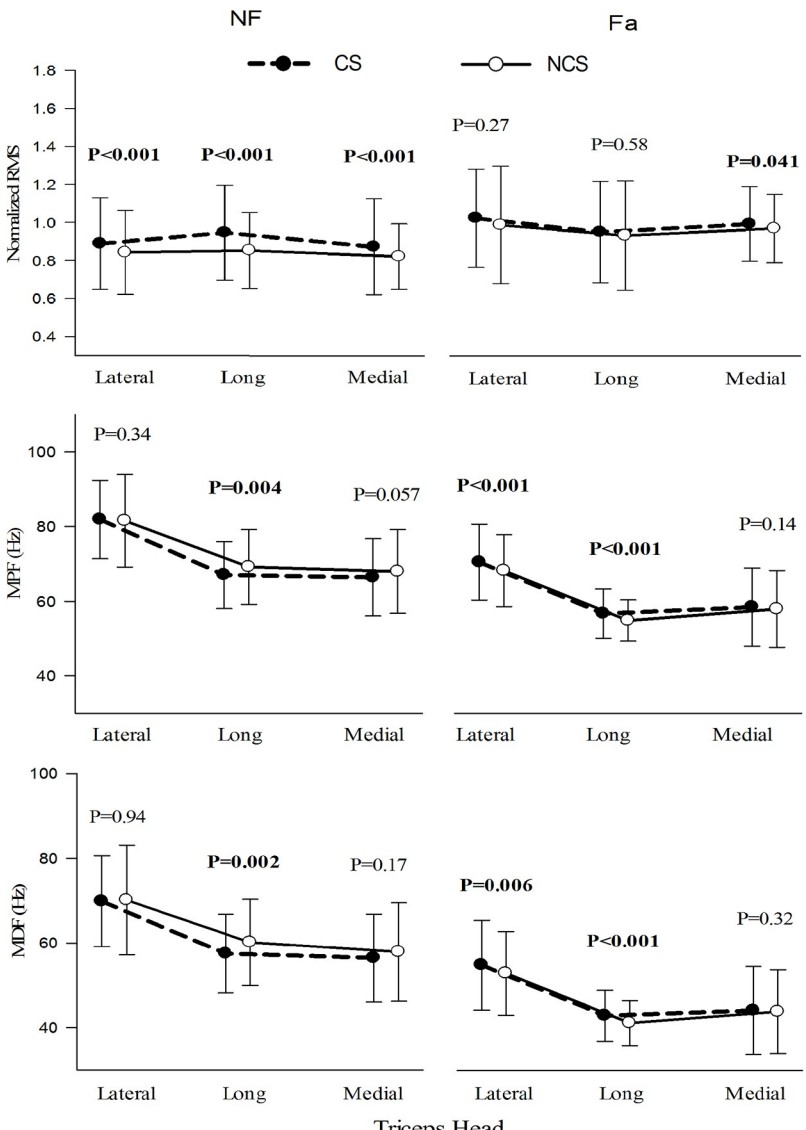

**Fig 5. RMS, MPF and MDF of the three heads of the TB under NF and Fa conditions during exercise with and without cognitive stress.**

conditions in both exercises. The absolute changes in the RMS, MPF and MDF from NCS to CS exercises under NF and Fa conditions are summarized in Table 5. The difference in RMS was more pronounced under the NF condition while the differences in MPF and MDF were comparable in both conditions.

**Table 2. Results of one-way ANOVA among the three heads under different conditions for RMS, MPF and MDF (P-values) and post hoc tests (a—lateral & long, b—long & medial, c—lateral & medial).**

| RMS 1 | RMS 2 | RMS 3 | RMS 4 | MPF 1 | MPF 2 | MPF 3 | MPF 4 | MDF 1 | MDF 2 | MDF 3 | MDF 4 |
|---|---|---|---|---|---|---|---|---|---|---|---|
| 0.27 | <0.001[a,c] | <0.001[b] | 0.002[a,c] | <0.001[a,c] | <0.001[a,b,c] | <0.001[a,c] | <0.001[a,b,c] | <0.001[a,c] | <0.001[a,b,c] | <0.001[a,c] | <0.001[a,b,c] |

*Bold font indicates statistically significant. 1: NF-NCS; 2: Fa-NCS; 3: NF-CS; 4: Fa-CS.

**Table 3. P-values for the main effects of cognitive stress and fatigue and their interaction on RMS, MPF and MDF for the three heads of the TB ($n = 25$).**

| | Lat | | | Lo | | | Med | | |
|---|---|---|---|---|---|---|---|---|---|
| | **RMS** | **MPF** | **MDF** | **RMS** | **MPF** | **MDF** | **RMS** | **MPF** | **MDF** |
| EXER | **0.003** | 0.084 | 0.148 | **0.008** | 0.897 | 0.456 | **0.003** | 0.824 | 0.755 |
| COND | **<0.001** | **<0.001** | **<0.001** | **<0.001** | **<0.001** | **<0.001** | **<0.001** | **<0.001** | **<0.001** |
| EXER × COND | **<0.001** | **0.047** | 0.061 | **0.003** | **<0.001** | **<0.001** | **<0.001** | **0.012** | 0.078 |

*Bold font indicates statistically significant.

**Table 4. ΔRMS(%), ΔMPF(%) and ΔMDF(%) from NF to Fa conditions, μ(SD).**

| | NCS | | | CS | | |
|---|---|---|---|---|---|---|
| | **Lat** | **Lo** | **Med** | **Lat** | **Lo** | **Med** |
| %ΔRMS | 11.01(20.6) | 3.35(30.07) | 6.28(25.25) | 22.33(26.81) | 8.63(28.07) | 21.11(17.33) |
| %ΔMPF | -28.22(17.76) | -35.64(15.34) | -28.2(14.86) | -30.98(17.8) | -44.3(19.53) | -34.25(23.7) |
| %ΔMDF | -29.34(21.2) | -33.72(19.3) | -26.6(17.95) | -33.16(21.5) | -44.79(24.8) | -32.3(26.3) |

**Table 5. ΔRMS, ΔMPF and ΔMDF from NCS to CS exercise, μ(SD).**

| | NF | | | Fa | | |
|---|---|---|---|---|---|---|
| | **Lat** | **Lo** | **Med** | **Lat** | **Lo** | **Med** |
| ΔRMS (mV) | 0.11(0.19) | 0.07(0.2) | 0.11(0.18) | -0.02(0.22) | 0.02(0.25) | -0.03(0.19) |
| ΔMPF (Hz) | 1.11(8.7) | -1.79(6.98) | -0.99(6.8) | 2.26(6.18) | 1.83(5.45) | 1.3(8.1) |
| ΔMDF (Hz) | 0.3(9.93) | -2.17(7.7) | -0.74(8.5) | 1.94(7.16) | 1.55(4.68) | 1.25(8.02) |

## Discussion

This study aimed to investigate the effect of CS on peripheral fatigue in the three heads of the TB. Global fatiguing effects were observed through the ET and NR, and the ROF, RMS, MPF and MDF for the three heads of the TB during the active phase under the different conditions were calculated and statistically compared. The RMS for all three heads increased in the presence of CS under NF conditions only, and the observed increase in the lateral and medial heads were comparable, confirming our hypothesis that CS would induce comparable increases in the RMS in synergist muscles. A higher percent decrease in the MPF and MDF for all three heads was observed in the presence of CS, which confirmed our second hypothesis.

The key findings of the current study indicate the functional variations in the sEMG patterns of the three heads of the TB in the presence of CS. Most previous studies examined the effect of CS on the trapezius muscle, which is most prone to be affected by CS [42,43,44,45]. In addition, some studies also included the arm, shoulder and neck muscles in their analysis of the effects of additional mental processing [11,12,27,46,47]. All of these studies have shown that the presence of CS increases neural and thus muscle activity compared with that found under neutral or non-mental-stress conditions. A previous study [48] investigated the deltoid and upper trapezius during isometric tasks with CS and found that CS reduced the endurance performance at 35% MVC but had no effect at 55% MVC. As observed in previous studies [32,48], CS might improve the endurance performance within a certain intensity range, but at other intensities, CS might have no or even a negative effect on exercise. At low exercise intensities, the extended time to task failure might cause central fatigue [49] and might thus negatively affect the ET, NR and ROF.

Attention diversion might partly explain the higher ET and NR observed during exercise with CS. As noted in a previous study [50], the ability of the CNS to decide whether to stop an exercise due to excessive load is reduced in the presence of CS, and this reduction causes the subject to work approximately 20% more until the CNS receives a sufficiently severe alert to stop the exercise. The dissociative focus observed in the presence of CS makes an individual more resistant to fatigue due to a reduction in fatigue perception because the individual is distracted from sensing fatigue [51]. The lower ROF observed in the current study might be responsible for the higher ET during the CS exercise, which concurs with a previous observation [32] of a higher ET during isometric elbow flexion performed at 50% MVC in the presence of CS. Taken together, it appears that the findings of this study concur with those in literature, suggesting similar behavior of ROF for both isometric and dynamic contractions during CS.

A higher RMS was found for all three heads during exercise in the presence of CS compared with NCS. Lateral and medial heads showed greater change in the RMS between the two exercises than for the long head, and this difference might be due to the different biomechanical roles of the TB heads. The long head, which is bi-articular, is more activated than other two mono-articular heads at a shoulder elevation angle of 0˚ [22], and maintained a constant force generating capacity across a wide range of elbow angles [20,21]. Because the exercises were performed at a shoulder elevation of 0˚, the long head is likely to show greater activity during both exercises, which would reduce the possibility of further changes in the activation levels due to the presence of CS. Two previous studies [52,53] revealed that the activation levels of the two other heads were higher but comparable. These observations appear to indicate that from a biomechanical viewpoint, the effects of CS in the lateral and medial heads are more pronounced than that in the long head. In addition, for all three heads, the change in the RMS between the NCS and CS exercises was more negligible under Fa compared with NF conditions. This observation could be due to the negligible effect of neural noise on muscles during fatigue, as observed in a previous study [16], and notes that a muscle exhibits a similar state of fatigue irrespective of the psychological state, as observed in another previous study [48].

The temporal and spectral parameters obtained for all three heads during exercise showed greater differences between NF and Fa conditions in the presence of CS, and this finding can be attributed to the increase in the exercise duration in the presence of CS. It is interesting to note that although the drop in MPF and MDF is higher under CS, their rates of drop (ROF) was found lower, albeit the higher energy cost during task execution under CS, as indicated by the higher sEMG activity. The reason for this could be the increased ET observed during the CS exercise. In addition, CS is known to induce neuro-motor noise in subjects, and an increase in neuro-motor noise causes an increase in kinematic variability. To suppress this variability, muscles tend to stiffen through agonist-antagonist co-activation. Thus, the observed increase in activity could be attributed to increased CS [18]. Although this finding was obtained for all three heads in our study, another research group [11] previously observed no change in the behavior of the lateral head of the TB with an increase in CS during finger tapping with a flexed elbow. These researchers claimed that their finding might be due to the co-contraction activity of the biceps brachii, which they found was more pronounced in the presence of CS. Although significant interactions between the two exercises and conditions were found for the RMS and MPF, the changes in the MPF and MDF obtained for both exercises between NF and Fa conditions were rather small, whereas the RMS exhibited greater changes. Hence, spectral parameters are less susceptible to cognitive stress than temporal parameters and are thus better approximators of peripheral muscle fatigue at different psychological states.

Comparable RMS values for the three heads were only obtained during the NCS exercise under NF conditions. In contrast, the three heads showed significant differences in the RMS,

MPF and MDF under all other conditions, further emphasizing the individual roles of each head. Collectively, the three heads continue to work individually in both conditions, NF and Fa. The analysis of synergist pairs revealed that the lateral-long and lateral-medial pairs showed consistent behavior in terms of the spectral parameters among all four exercise-condition cases, namely, NF-NCS, Fa-NCS, NF-CS and Fa-CS. Although the responses of the long-medial head pair during both NCS and CS exercises under Fa conditions were consistent with those of the two other synergist pairs, the opposite behavior was observed under NF conditions. This finding could be related to the fact that the long head, which is bi-articular, experiences increased changes in length and joint angles compared with its mono-articular counterparts, the lateral and medial heads [52]. The behavior of the lateral-medial synergist pair was consistent because both heads are mono-articular, and hence, external and internal factors exert similar effects on these heads. In contrast, the lateral-long synergist pair function well in unison because both are large muscles with similar numbers of muscle fibers [54]. However, the behavior of the long-medial pair cannot be ascertained because the two muscles belonging to this pair do not share such characteristics (i.e., one of the heads is not mono-articular, and the two heads have different muscle sizes).

Muscle fatigue consists of two components, peripheral and central fatigue. Because the central mechanisms were not examined in tandem with neuromuscular behavior, the possibility that it plays a role in the manifestation of peripheral fatigue cannot be excluded. The inclusion of such an analysis might provide more convincing and accurate findings that provide further insights on the mechanisms and relationships between CS and peripheral fatigue. The effects of crosstalk among the three heads of the TB, particularly between the medial and lateral heads cannot be outright neglected. By adhering to the recommendations in [55] precautionary measures were taken, and the researchers are confident that the effect of crosstalk is negligible.

## Conclusion

The presence of CS decreases the ROF and increases the ET and NR. For all three heads, larger differences were observed in the normalized RMS between the NCS and CS exercises under NF compared with Fa conditions. Temporal parameters were found to be better approximators of CS, whereas spectral parameters were more resistant to the effect of CS. In addition to previous results for individual heads, the findings obtained in this study further affirm that the three TB heads work independently under fatiguing conditions and in the presence of CS. However, an analysis of the various pairs of TB heads revealed that the behavior of the non-synergist head pair under fatiguing conditions was similar to those of the synergist head pairs. These results could help researchers obtain a more in-depth understanding on the functioning of the TB and thus can potentially be used in clinical applications for prosthetic control or targeted sports training. Furthermore, the effects of CS on peripheral muscle fatigue can improve the understanding of the condition of an individual during training or rehabilitation. Future work may also consider using the attention diversion technique to induce CS in athletes, who are usually more focused during task execution.

## Acknowledgments

The authors would like to acknowledge Universiti Teknikal Malaysia Melaka (UTeM) for providing the research facilities. The authors would also like to thank the physicians that participated in this study, the Director General of Health Malaysia for giving permission to publish this paper and the Medical Research and Ethics Committee (MREC) of Malaysia for providing ethical approval to collect the data used in this study.

## Author Contributions

**Conceptualization:** Jawad Hussain, Kenneth Sundaraj.

**Data curation:** Jawad Hussain.

**Formal analysis:** Jawad Hussain.

**Investigation:** Jawad Hussain.

**Methodology:** Jawad Hussain, Kenneth Sundaraj.

**Project administration:** Kenneth Sundaraj.

**Supervision:** Kenneth Sundaraj.

**Validation:** Jawad Hussain.

**Visualization:** Indra Devi Subramaniam.

**Writing – original draft:** Jawad Hussain, Indra Devi Subramaniam.

**Writing – review & editing:** Kenneth Sundaraj, Indra Devi Subramaniam.

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
