## [Decision Letter · Decision Letter 0]

5 Aug 2019

PONE-D-19-18253

Cognitive Stress Changes the Attributes of the Three Heads of the Triceps Brachii during Fatigue

PLOS ONE

Dear Mr. Hussain,

Thank you for submitting your manuscript to PLOS ONE. After careful consideration, we feel that it has merit but does not fully meet PLOS ONE’s publication criteria as it currently stands. Therefore, we invite you to submit a revised version of the manuscript that addresses the points raised during the review process.

We would appreciate receiving your revised manuscript by Sep 19 2019 11:59PM. To enhance the reproducibility of your results, we recommend that if applicable you deposit your laboratory protocols in protocols.io, where a protocol can be assigned its own identifier (DOI) such that it can be cited independently in the future. For instructions see: http://journals.plos.org/plosone/s/submission-guidelines#loc-laboratory-protocols

We look forward to receiving your revised manuscript.

Kind regards,

Nizam Uddin Ahamed, PhD

Academic Editor

PLOS ONE

3. We note that Figure 1 includes an image of a participant in the study.

Reviewers' comments:

Reviewer's Responses to Questions

**Comments to the Author**

1. Is the manuscript technically sound, and do the data support the conclusions?

Reviewer #1: Yes

Reviewer #2: Partly

2. Has the statistical analysis been performed appropriately and rigorously? 

Reviewer #1: Yes

Reviewer #2: Yes

3. Have the authors made all data underlying the findings in their manuscript fully available?

Reviewer #1: Yes

Reviewer #2: No

4. Is the manuscript presented in an intelligible fashion and written in standard English?

Reviewer #1: Yes

Reviewer #2: Yes

5. Review Comments to the Author

Reviewer #1: The purpose of the study brings up an interesting question. By assessing EMG from each individual head from the triceps brachii, the current study was wanting to assess peripheral fatigue, and muscle firing patterns of each of the individual muscles with two different treatment conditions. The present study was trying to assess individual specific muscle responses with and without cognitive stresses. While the research question is relevant, interesting, and could add to the literature (specifically including synergist muscles), there are few clarifications required prior to publication of the article.

General comments:

Please remain consistent with the terminology throughout the article, such as cognitive stress and mental load were used interchangeably. It would be better if the authors replace ML (for mental load) with CS (cognitive stress) as CS is appearing in the title. Though on P3L7, authors mentioned “… mental load induced by cognitive stress.”, the use of CS is recommended for this particular work.

Further, peripheral fatigue has been closely related to a slowing of the sarcolemma and t-tubule conduction velocity (Zhou 1996 and Fitts 2006). By having participants perform dynamic contractions at 45% 1-RM to failure, it is plausible that the participant may get 55% fatigued compared to their baseline strength. Iguchi et al. 2008, suggests that fatigue that induces at least 35% or greater could induce low frequency fatigue (LFF), which could delay recovery responses for a prolonged period of time.

Particular comments:

P2L20,” …trapezius, neck and shoulder muscles…” may be replaced by “…neck, trapezius and shoulder muscles…” (note the order of the muscles).

P3L2, “amount of cognitive stress.” This phrase suggests that cognitive stress is quantifiable! Is it the case? Please clarify.

P3L4, mental may be replaced by cognitive.

P3L6, It should be specified that TB is involved in horizontal arm abduction.

P3L7-9, why TB activity was overshadowed? Due to task performed? Please clarify in the manuscript.

P3L12, why there is the word “additional”. Does the subjects were already suffering from cognitive stress?

P5L1,” …23.8±3.6…” may be replaced by 23.8 (3.6) etc.

P5L10, The skin should always be shaved (as per SENIAM recommendations). Why was the skin only shaved as needed? How was “as needed” defined?

P5L9, How the authors identified medial head of the TB? There is no recommendation regarding medial head in SENIAM!

P5L18, This sampling rate was on the low end for surface EMG. It is important that you state the specifications of the Shimmer system. What were the analog high and low pass cutoff frequencies? If the high pass was greater than 499 Hz, it is possible that you got some aliasing in your signal due to the relatively low sampling rate. I can't find any information on this amplifier, even on the RealTime website. I did find information for the Shimmer 3.0, but it does not indicate the analog filtering used, only that the bandwidth is 8.5 kHz

Methodology section: At which speed 1RM test and rest of the experiment was performed? Did you measure the actual angular velocity during the contractions?

P6L10, How the proper posture was ensured? Was an assistant dedicated for this purpose or was that done through some equipment?

Was control of movement only experimenter’s confirmation? Did authors measure joint angles by electrical goniometers or motion capture?

Joint angles when surface EMG was sampled were not controlled.

P6L12, between which trials? It may be written as “between 1RM trials”.

As suggested in the general comments, NML and ML may be replaced with NCS and CS.

P5L22, Why was the study all done on one day? Why was the study not spread out over 3 days? Day 1 – familiarization and 1-RM, day 2 one condition, day 3 the other condition.

P7L8,9, the sentence, “The tempo of the exercise was maintained at subject’s ease.” is not enough. Authors should mention WHY the tempo of exercise was maintained at subject’s ease.

P7L19, What did you use to normalize the RMS. This is not clear from the description at Line P7L19. How exactly did you determine the MVC from the 1 RM contraction? Was it the highest value? The average?

P7L21-24, please clarify the sentence. Some information seems overwritten. Please clarify the methodology for the detection of active phase.

P8L2, Please correct. It seems “for all the segments” is missing.

As you used parametric analyses, did your data satisfy the assumption of a normal distribution?

The reasons that effects of mental stress are different among the synergistic muscles are not stated. Please explain them with references.

Authors should state how the EMG data was sampled for further analysis (data window) more than authors described in P7 L19-24 and Fig. 2. During dynamic movements, surface EMG signal is strongly influenced by non-physiological factors, i.e., distance between electrode and innervation zone etc. So, we need to control the joint angle when surface EMG is sampled during dynamic movement. Only visual inspection, the data from non-fatigue and fatigue periods or the data from with and without mental stress may be sampled from different joint angle range. If you sampled surface EMG from different joint range, this difference in surface EMG cannot be explained by the effect of fatigue or mental stress.

What are the units of rate of fatigue?

what was the purpose of calculating both MPF and MDF? Was there unique information you expected to uncover from each? It does not appear so from the Results and Discussion and you should consider only presenting one, or justifying why it is important to present both.

Although the result section is written fairly detailed, it needs some modifications. The independent and dependent variables need to be identified and reported properly. For example, intensity may have a significant effect on RMS, but it is not RMS that is significant.

I am not sure it is necessary to be so specific with p values that are so far below zero. (eg. 8.15E-41). It just clutters up the table and the specificity does not add anything. Consider replacing these with <0.001 to be consistent with the 3 decimal places for the other p values.

Reviewer #2: Cognitive Stress Changes the Attributes of the Three Heads of the Triceps Brachii during Fatigue

Manuscript Number: PONE-D-19-18253

This study aims to examine the effect of cognitive stress on the activity behavior of the three heads of the triceps brachii (TB) during dynamic fatiguing exercise (triceps push-down exercise at 45% of 1RM). The authors found that in the presence of cognitive stress, the rate of fatigue (ROF) for all the three heads of the TB demonstrated no significant differences, whereas endurance time and number of repetitions increased. Significant differences were found in the RMS of the EMG activity of all three heads of TB between the two exercises (with and w/o the cognitive load) under no muscle fatigue condition, whereas no differences were found under fatigue condition. Only the long head of TB demonstrated sig. differences of MPF and MDF between the exercises with and without mental load, regardless of fatigue condition. The authors concluded that spectral parameters acquired from the three heads of TB were found to be more robust towards the effect of cognitive stress on the fatiguing TB.

The MS in general is well-written, but there are parts that need improvement so as to become more clear and more easily understood, e.g. the results sections of abstract and MS (see also specific comments below). It is an interesting topic, but I have concerns whether the used methodology is appropriate for drawing concrete conclusions about neural noise and peripheral fatigue.

Specific comments:

Abstract:

Page 8, line 19: Please rephrase the sentence in order to become easier to follow: “Similarly, the RMS was statistically significant between the two exercises under NF conditions (P<0.05), whereas it was non-significant (P>0.05) under Fa conditions”

Introduction:

Page 10, Line 7: Please specify compared to what is the TB “more susceptible to the effects of cognitive stress”.

Page 10, Line 24: Please explain, why isometric contractions require focus on postural control. For example, when the isometric contraction is performed at a sitting position and on an isokinetic dynamometer, on which the person is well secured, no focus on postural control is required. Maybe the authors just want to justify why they preferred dynamic instead of isometric contraction, so it would be better to rephrase accordingly.

Methodology:

Experimental procedure:

• A figure, depicting the experimental procedure would be helpful for understanding the experiment set up.

• Have you checked if the mental load exercise has an effect on the execution of 1RM? In other words, whether the maximum lifted weight is affected by mental loading? It is possible that mental load may influence the force generation capacity, and thus although the fatiguing tasks are performed at the same weight-lift, this same weight-lift may correspond to different work load. Please comment on that.

Page 13/ Line 17: While you define task failure later in the MS, it would be better for the reader to provide the definition already at this point.

Page 14/ Line 4: Have you checked the recovery? Had the subjects the same 1RM before starting the second fatiguing task?

Page 14/ Line 10: Have you checked range of motion during the repetitions?

Statistical analysis:

• It would be interesting to see if there is any interaction between the effect of mental load and fatigue condition. This could be examined by using statistical tests like two-way ANOVA, with the one factor being the mental load and the second fatigue condition.

• What statistical test was used for checking the differences of the ROF, endurance time and number of repetitions between the two fatiguing exercises?

Page 15/ Line 11: Please change “spehricity” into “sphericity”

Results:

Page 15/ Line 22: Please address also the mean and SD values of the endurance time and number of reps.

Figure 3: The statistically significant differences are not shown in the figure. This would make the results demonstration easier to understand. Furthermore, why is there a shift between ML and NML in figure?

The results session is complicatedly demonstrated. It should be made more clear to the reader which are the most important results.

Discussion:

Page 18/ Line 6: Since the central mechanisms were not examined, but only the neuromuscular behavior by means of EMG, the authors should explain the accuracy of their conclusions on peripheral fatigue. E.g. how can it be excluded that no central mechanisms were involved in the changes of neuromuscular activity?

Page 19/Line 6: The authors state that: “The ROF of all three heads was higher in the NML exercise compared with the ML exercise”, however no statistically significant differences were found. Please rephrase.

Page 19/Line 11: The authors address that RMS showed greater variability in the ML exercises between NF and Fa conditions, and this variability can be explained by neural noise. Nevertheless, neither in methodology session nor in the results session there is any reference on the RMS variability; i.e. how was variability assessed and what were its values (only RMS differences between trials and conditions are shown). Furthermore, please provide reference on how RMS variability is related to neural noise.

Page 20, Line 1: It is not so clear how from the references [7,40] the addressed conclusion is made, i.e. that larger muscles show decreased variability in joint movements in the presence of cognitive stress, which indicates that these muscles can more effectively handle neuro-motor noise than smaller muscles. Firstly [40], in their study do not examine cognitive stress. There is no reference on cognitive stress in their paper. Secondly, [7] report as possible explanations for the differences in muscle activity following mental load (in terms of amplified memory processing) between triceps and biceps brachii, their anatomical difference (biarticular vs, monoarticular) and functional proximity to the shoulder, but not their size. Please provide appropriate reference to support the finding that larger muscles show decreased variability in joint movements in the presence of cognitive stress.

Page 20/Line 18: I would like a better justification, on how the results are related to neuro-motor noise. How can conclusions be made on the role of neuro-motor noise, from the differences between the RMS values?

6. PLOS authors have the option to publish the peer review history of their article (what does this mean?). If published, this will include your full peer review and any attached files.

Reviewer #1: Yes: Rajkumar Palaniappan

Reviewer #2: No

---

## [Author Response · Author response to Decision Letter 0]

14 Sep 2019

The manuscript has been formatted as per PLOS ONE’s style requirements. About the availability of the data, it was not mentioned explicitly in our ethical approval document that the data can be shared publicly. The data may be shared upon request to the corresponding author. We have added a statement in the revised manuscript regarding Figure 1.

Detailed response to the reviewers' comments has been attached.

---

## [Decision Letter · Decision Letter 1]

9 Oct 2019

PONE-D-19-18253R1

Cognitive stress changes the attributes of the three heads of the triceps brachii during fatigue

PLOS ONE

Dear Mr. Hussain,

Thank you for submitting your manuscript to PLOS ONE. After careful consideration, we feel that it has merit but does not fully meet PLOS ONE’s publication criteria as it currently stands. Therefore, we invite you to submit a revised version of the manuscript that addresses the points raised during the review process.

We would appreciate receiving your revised manuscript by Nov 23 2019 11:59PM. To enhance the reproducibility of your results, we recommend that if applicable you deposit your laboratory protocols in protocols.io, where a protocol can be assigned its own identifier (DOI) such that it can be cited independently in the future. For instructions see: http://journals.plos.org/plosone/s/submission-guidelines#loc-laboratory-protocols

We look forward to receiving your revised manuscript.

Kind regards,

Nizam Uddin Ahamed, PhD

Academic Editor

PLOS ONE

Additional Editor Comments (if provided):

Thank you for carefully revised the manuscript based on the reviewer comments and addressed all issues raised with respect to the first version in a satisfactory way. However, please have the entire manuscript edited for English expression by a native speaker of English who has knowledge of your field. Language imprudent is required; avoid the first-person writing style.

There are some comments those need to be addressed before further processing:

1. One important key word “Muscle” is missing in the title, therefore I would recommend putting this word either with triceps brachii “....triceps brachii muscle...” or just before the word fatigue, “…..triceps brachii during Muscle fatigue”

2. Define the acronym(s), first time that they are used; such as in Abstract, iRM, RMS, MPF and MDF. Please check the entire manuscript.

3. The keywords need to arrange based on alphabetical order.

4. In the Introduction, first paragraph: only two references (one is very old), are not enough to support the statement. Specially the external factor “Temperature”,. Please add relevant and recent articles. In the same paragraph, instead of writing “[1] revealed that…”, please put the author name(s). Similarly, please add reference(s) to support your first two statements in the second paragraph.

5. In the Introduction, “The triceps brachii, as the largest arm muscle…’, is there any right reference other than your own article to support this? Because, Biceps Brachii muscle is recognized as largest and right place to place EMG electrode as per SENIAM.

6. Please revise the sentence “Because the TB is close to the CNS..”, because, facial muscle, eye muscle etc. are the close to CNS.

Reviewers' comments:

Reviewer's Responses to Questions

**Comments to the Author**

1. If the authors have adequately addressed your comments raised in a previous round of review and you feel that this manuscript is now acceptable for publication, you may indicate that here to bypass the “Comments to the Author” section, enter your conflict of interest statement in the “Confidential to Editor” section, and submit your "Accept" recommendation.

Reviewer #1: All comments have been addressed

Reviewer #2: (No Response)

2. Is the manuscript technically sound, and do the data support the conclusions?

Reviewer #1: Yes

Reviewer #2: Partly

3. Has the statistical analysis been performed appropriately and rigorously? 

Reviewer #1: Yes

Reviewer #2: Yes

4. Have the authors made all data underlying the findings in their manuscript fully available?

Reviewer #1: Yes

Reviewer #2: No

5. Is the manuscript presented in an intelligible fashion and written in standard English?

Reviewer #1: Yes

Reviewer #2: Yes

6. Review Comments to the Author

Reviewer #1: "Cognitive stress changes the attributes of the three heads of the triceps brachii during fatigue "

The authors have addressed all the comments in the revised manuscript. The manuscript can be accepted in the present form.

Reviewer #2: Cognitive Stress Changes the Attributes of the Three Heads of the Triceps Brachii during Fatigue

Manuscript Number: PONE-D-19-18253

The authors replied and tried to address all my comments. However, I still have some concerns, especially on the fact on how the results are related to neuro-motor noise. When conclusions for motor units behavior are drawn from bipolar surface EMG, this should be done very carefully. The conclusion that the greater RMS increase during the CS exercise compared to the NCS could be due to neuro-motor noise, still needs justification. It is possible that an increase in EMG activity, can be explained due to the increased antagonist-agonist co-contraction, which can be an indication of higher neuro-motor noise in order to counteract the kinematic variability [as stated by “The effects of workplace stressors on muscle activity in the neck shoulder and forearm muscles during computer work: a systematic review and meta-analysis”], but there are many other factors as well. For example, it is possible that the greater duration of the exercise under CS may explain the greater RMS increase. Furthermore, the authors have not investigated neither co-contraction nor kinematics variability, which could be indicators for motor noise existence, so that they can argue that motor noise is the factor explaining RMS increase.

Specific comments:

Methods:

1. In the previous review I commented that a figure depicting the experimental procedure would be helpful, meaning the temporal set up, e.g. first warming up then 2 min rest, 3 RM (the between intervals), etc. I believe that the words “experimental set up” confused the authors, and added a figure depicting the spatial set up of the experiment. Nevertheless, this is also described in figure 1, so a suggestion could be to keep either figure 1 or figure 2 and add another figure depicting the temporal experimental procedure.

2. Page 7/Line 16-18: At the present study EMG parameters (RMS, MDF and MPF) were compared between different conditions. In order to investigate the effect of the different conditions, motion should be the same, i.e. constant ROM at constant velocity, since these two parameters influence the EMG signal. Thus, I disagree with the statement that the present study did not require the joint angles to be measured or controlled during sEMG data acquisition as data was required throughout the entire ROM.

3. Page 9/Line 8-10: I have some concerns on how accurate the duration of one repetition can be defined from EMG signal. There is an activation of TB during elbow flexion (eccentric phase) and also during elbow extension (concentric phase) with a small silent window at the end of each repetition. The duration of this silent time window could be fatigue depended.

Results:

4. Results session continuous to be complicatedly demonstrated. For example, it is mentioned that “Under the NF condition, all three heads were significantly different (p<0.05) between the NCS and CS exercises for RMS, while only the long head was statistically significant for MPF and MDF.” But there is no information on which variable is higher. Furthermore, in figure 5, 6 and 7 the statistically significant differences are not indicated.

Discussion:

5. It would be easier for the reader to follow and understand the main findings of the study, if at the beginning of the discussion the main findings are described and whether these findings confirm or reject the hypotheses (a. that the RMS activity of the lateral and long heads would increase in the presence of cognitive stress because these are larger muscles, whereas that of medial head would remain and b. that the values of MPF and MDF for the lateral and long heads would experience a relatively higher decrease in the presence of cognitive stress compared with those for the medial head). Furthermore, at the introduction is stated that the rate of fatigue (ROF) is expected to be higher in the presence of cognitive stress than under normal conditions. However, this expectation was not confirmed from the results. A restructure of the discussion where the justification, of the confirmation or rejection of the hypotheses should be done in order to improve discussion clarity.

6. There is a significant interaction found between mental load and fatigue development in the RMS and in all the heads of TB for RMS and MPF, which is not discussed in the discussion part.

7. PLOS authors have the option to publish the peer review history of their article (what does this mean?). If published, this will include your full peer review and any attached files.

Reviewer #2: No

---

## [Author Response · Author response to Decision Letter 1]

7 Nov 2019

Response to Editor and Reviewer comments are in the file "Response to Reviewers.docx"

---

## [Decision Letter · Decision Letter 2]

21 Nov 2019

PONE-D-19-18253R2

Cognitive stress changes the attributes of the three heads of the triceps brachii during muscle fatigue

PLOS ONE

Dear Mr. Hussain,

Thank you for submitting your manuscript to PLOS ONE. After careful consideration, we feel that it has merit but does not fully meet PLOS ONE’s publication criteria as it currently stands. Therefore, we invite you to submit a revised version of the manuscript that addresses the points raised during the review process.

We would appreciate receiving your revised manuscript by Jan 05 2020 11:59PM. To enhance the reproducibility of your results, we recommend that if applicable you deposit your laboratory protocols in protocols.io, where a protocol can be assigned its own identifier (DOI) such that it can be cited independently in the future. For instructions see: http://journals.plos.org/plosone/s/submission-guidelines#loc-laboratory-protocols

We look forward to receiving your revised manuscript.

Kind regards,

Nizam Uddin Ahamed, PhD

Academic Editor

PLOS ONE

Reviewers' comments:

Reviewer's Responses to Questions

**Comments to the Author**

1. If the authors have adequately addressed your comments raised in a previous round of review and you feel that this manuscript is now acceptable for publication, you may indicate that here to bypass the “Comments to the Author” section, enter your conflict of interest statement in the “Confidential to Editor” section, and submit your "Accept" recommendation.

Reviewer #2: (No Response)

2. Is the manuscript technically sound, and do the data support the conclusions?

Reviewer #2: Yes

3. Has the statistical analysis been performed appropriately and rigorously? 

Reviewer #2: Yes

4. Have the authors made all data underlying the findings in their manuscript fully available?

Reviewer #2: No

5. Is the manuscript presented in an intelligible fashion and written in standard English?

Reviewer #2: Yes

6. Review Comments to the Author

Reviewer #2: Cognitive Stress Changes the Attributes of the Three Heads of the Triceps Brachii during Fatigue

Manuscript Number: PONE-D-19-18253

The authors replied and tried to address all my comments. However, I still have some concerns mainly in the discussion part.

• In the revised discussion the authors added that “spectral parameters are less susceptible to neuro-motor noise than temporal parameters and are thus better approximators of peripheral muscle fatigue at different psychological states.” Please elaborate on how this conclusion can be drawn from the results of the present study since the presented methodology does not measure neural noise (cf. relevant comment of previous review).

• In the present study, the two hypotheses are confirmed while the third is rejected. In the introduction part the third hypothesis is based on the second: “…the presence of CS likely induces a relatively greater decrease in the MPF and MDF and thus a greater ROF than that observed under normal condition”. Please elaborate why, while the second hypothesis is confirmed, the third in rejected, clearly in the discussion part.

Specific comments

Discussion:

• Page 14/Line15-17: Please provide a literature reference for the statement that: “At low exercise intensities, the extended time to task failure might cause central fatigue and might thus negatively affect the ET, NR and ROF.”

Conclusion:

• Page 17/Line 11: Please change the word “but” to “and”, since it is reasonable a reduced ROF to lead to an increase in the ET and NR.

• Page 17/Line 11: Please cite on what the impact of CS was greater under NF conditions.

7. PLOS authors have the option to publish the peer review history of their article (what does this mean?). If published, this will include your full peer review and any attached files.

Reviewer #2: No

---

## [Author Response · Author response to Decision Letter 2]

25 Nov 2019

A file has been attached containing our response to the comments of the reviewer.

---

## [Decision Letter · Decision Letter 3]

10 Dec 2019

PONE-D-19-18253R3

Cognitive stress changes the attributes of the three heads of the triceps brachii during muscle fatigue

PLOS ONE

Dear Mr. Hussain,

Thank you for submitting your manuscript to PLOS ONE. After careful consideration, we feel that it has merit but does not fully meet PLOS ONE’s publication criteria as it currently stands. Therefore, we invite you to submit a revised version of the manuscript that addresses the points raised during the review process.

We would appreciate receiving your revised manuscript by Jan 24 2020 11:59PM. To enhance the reproducibility of your results, we recommend that if applicable you deposit your laboratory protocols in protocols.io, where a protocol can be assigned its own identifier (DOI) such that it can be cited independently in the future. For instructions see: http://journals.plos.org/plosone/s/submission-guidelines#loc-laboratory-protocols

We look forward to receiving your revised manuscript.

Kind regards,

Nizam Uddin Ahamed, PhD

Academic Editor

PLOS ONE

Reviewers' comments:

Reviewer's Responses to Questions

**Comments to the Author**

1. If the authors have adequately addressed your comments raised in a previous round of review and you feel that this manuscript is now acceptable for publication, you may indicate that here to bypass the “Comments to the Author” section, enter your conflict of interest statement in the “Confidential to Editor” section, and submit your "Accept" recommendation.

Reviewer #2: (No Response)

2. Is the manuscript technically sound, and do the data support the conclusions?

Reviewer #2: Yes

3. Has the statistical analysis been performed appropriately and rigorously? 

Reviewer #2: Yes

4. Have the authors made all data underlying the findings in their manuscript fully available?

Reviewer #2: (No Response)

5. Is the manuscript presented in an intelligible fashion and written in standard English?

Reviewer #2: Yes

6. Review Comments to the Author

Reviewer #2: The authors replied and tried to address all my comments. However, in the revised manuscript new issues in the discussion part arise.

• In the introduction part the second hypothesis, i.e. the presence of CS would induce a relatively greater decrease in the MPF and MDF is based on the assumption that there would be greater muscle activity during exercise. Indeed, the authors found greater RMS values under CS. Interestingly the ROF is lower and the duration is greater under CS, albeit the higher energy cost during task execution under CS, as indicated by the higher EMG activity. Is it possible that the differences in task duration may have influenced the MPF and MDF reduction between the different conditions (CS vs NCS)? This issue should be explained/commented in the discussion section of the paper.

• Page 14/ Line 23-25: The authors state that “The higher ET observed in our study might also be responsible for the lower ROF calculated during exercise with CS”. As the higher ET is the result of the art that the neuromuscular system is operated and not vice versa, it is more reasonable to write that the lower ROF observed in the study might be responsible for the higher ET observed during exercise with CS. Please revise the discussion accordingly.

7. PLOS authors have the option to publish the peer review history of their article (what does this mean?). If published, this will include your full peer review and any attached files.

Reviewer #2: No

---

## [Author Response · Author response to Decision Letter 3]

18 Dec 2019

Response to reviewer's comments is attached.

---

## [Decision Letter · Decision Letter 4]

8 Jan 2020

Cognitive stress changes the attributes of the three heads of the triceps brachii during muscle fatigue

PONE-D-19-18253R4

Dear Dr. Hussain,

We are pleased to inform you that your manuscript has been judged scientifically suitable for publication and will be formally accepted for publication once it complies with all outstanding technical requirements.

With kind regards,

Nizam Uddin Ahamed, PhD

Academic Editor

PLOS ONE

Additional Editor Comments (optional):

Reviewers' comments:

Reviewer's Responses to Questions

**Comments to the Author**

1. If the authors have adequately addressed your comments raised in a previous round of review and you feel that this manuscript is now acceptable for publication, you may indicate that here to bypass the “Comments to the Author” section, enter your conflict of interest statement in the “Confidential to Editor” section, and submit your "Accept" recommendation.

Reviewer #2: All comments have been addressed

2. Is the manuscript technically sound, and do the data support the conclusions?

Reviewer #2: (No Response)

3. Has the statistical analysis been performed appropriately and rigorously? 

Reviewer #2: (No Response)

4. Have the authors made all data underlying the findings in their manuscript fully available?

Reviewer #2: (No Response)

5. Is the manuscript presented in an intelligible fashion and written in standard English?

Reviewer #2: (No Response)

6. Review Comments to the Author

Reviewer #2: (No Response)

7. PLOS authors have the option to publish the peer review history of their article (what does this mean?). If published, this will include your full peer review and any attached files.

Reviewer #2: No

---

## [Editor Report · Acceptance letter]

15 Jan 2020

PONE-D-19-18253R4 

Cognitive stress changes the attributes of the three heads of the triceps brachii during muscle fatigue 

Dear Dr. Hussain:

I am pleased to inform you that your manuscript has been deemed suitable for publication in PLOS ONE. Congratulations! Your manuscript is now with our production department. 

With kind regards,

on behalf of

Dr. Nizam Uddin Ahamed 

Academic Editor

PLOS ONE